# Enhancing Dissolution Rate and Antibacterial Efficiency of Azithromycin through Drug-Drug Cocrystals with Paracetamol

**DOI:** 10.3390/antibiotics10080939

**Published:** 2021-08-04

**Authors:** Noor Ul Islam, Ezzat Khan, Muhammad Naveed Umar, Attaullah Shah, Muhammad Zahoor, Riaz Ullah, Ahmed Bari

**Affiliations:** 1Department of Chemistry, University of Malakand, Chakdara 18800, Pakistan; nooruomchem@gmail.com (N.U.I.); ekhan@uom.edu.pk (E.K.); 2Department of Chemistry, College of Science, University of Bahrain, Sakhir 32038, Bahrain; 3Jacobs University School of Life Sciences and Chemistry, Campus Ring 1, 28759 Bremen, Germany; m.naveedumar@uom.edu.pk; 4Pakistan Institute of Engineering and Applied Sciences, National Institute of Lasers and Optronics College (NILOP-C, PIEAS), Islamabad 44000, Pakistan; attashah168@gmail.com; 5Department of Biochemistry, University of Malakand, Chakdara 18800, Pakistan; 6Department of Pharmacognosy, College of Pharmacy, King Saud University, Riyadh 11451, Saudi Arabia; rullah@ksu.edu.sa; 7Department of Pharmaceutical Chemistry, College of Pharmacy, King Saud University, Riyadh 11451, Saudi Arabia; abari@ksu.edu.sa

**Keywords:** antibacterial agents, azithromycin, cocrystallization, dissolution studies, paracetamol

## Abstract

Cocrystallization is a promising approach to alter physicochemical properties of active pharmaceutical ingredients (hereafter abbreviated as APIs) bearing poor profile. Nowadays pharmaceutical industries are focused on preparing drug-drug cocrystals of APIs that are often prescribed in combination therapies by physicians. Physicians normally prescribe antibiotic with an analgesic/antipyretic drug to combat several ailments in a better and more efficient way. In this work, azithromycin (AZT) and paracetamol (PCM) cocrystals were prepared in 1:1 molar ratio using slow solvent evaporation method. The cocrystals were characterized by Fourier transform infrared (FTIR), Raman spectroscopy, powder X-ray diffraction (PXRD), differential scanning calorimeter (DSC), thermo gravimetric analysis (TGA) and high-performance liquid chromatography (HPLC). Vibrational spectroscopy and DSC confirmed that both APIs interact physically and showed chemical compatibility, while PXRD pattern of the starting material and products revealed that cocrystal have in a unique crystalline phase. The degree of hydration was confirmed by TGA analysis and result indicates monohydrate cocrystal formation. The HPLC analysis confirmed equimolar ratio of AZT:PCM in the cocrystal. The in vitro dissolution rate, saturation solubility, and antimicrobial activity were evaluated for AZT dihydrate and the resulting cocrystals. The cocrystals exhibited better dissolution rate, solubility and enhanced biological activities.

## 1. Introduction

Recently, researchers have focused on preparation of drug-drug cocrystals (DDCs) for commercial purposes. This approach is more practical as physician describes combination of different drugs to get far reaching results and if such combination is converted into cocrystals it will not only reduce the compliance of the patients but will be economic as well. Currently, limited literature is available to describe DDCs. The DDC of meloxicam and aspirin cocrystal has been developed with a better bioavailability and therapeutic concentration in blood plasma as compared to meloxicam [1]. Better dissolution [2] and several other physical properties have been improved for some drugs with some limitations [3]. The techniques of cocrystallization have been very successful in HIV related drugs [4], combination drugs [5], tuberculosis [6], effective antibacterial drugs [7]. Patient compliance, solubility, dissolution rate, bioavailability, and stability of at least one component are improved through DDCs [8]. Though DDCs syntheses are a viable strategy to modulate properties of a drug to resolve the associated problems with conventional combination drugs therapy [9]. However, it may have a dose proportionality issue in designing such type of cocrystals. As the molar ratio of drugs is fixed, cocrystal formation occurs according to the stoichiometric rule, although the dose of each drug varies depending on the patient’s age, race or ethnicity and clinical indications [8]. 

Azithromycin (AZT, shown in Figure 1) is a semisynthetic macrolide derivative of erythromycin A found as monohydrate (MH), dihydrate (DH), semi-hydrate solvates [10,11,12] and pseudopolymorphic forms [13]. However, the MH and DH crystalline forms are more stable compared to other forms [13,14]. The MH form is used less due hygroscopic properties while the DH form is preferably used for therapeutic applications [13]. AZT in biopharmaceutical classification system (BCS) is class-II (low solubility/high permeability) API [15]. It is used in the treatment of several pediatric and adult infections [16,17,18], such as lower and upper respiratory tract infection, bronchitis, laryngitis, pneumonia and middle ear infections [19]. Additionally, it is also an effective therapeutic antibiotic used in the treatment of other diseases like tonsillitis, sexually transmitted diseases and skin infections [20]. When the drug is administered orally, its solubility in physiological fluid is very poor, which accounts for low bioavailability [21]. The drug, AZT, absorbs quickly, exhibiting excellent bioavailability once its solubility problem is resolved. Therefore, to achieve good pharmacological results, the solubility of the drug must be enhanced for improved absorption and bioavailability [15]. Furthermore, the drug exhibits bitter taste [22] and has been found to be degraded photocatalytically [23]. A number of approaches like nanoparticles [24,25], nanosuspension [26], microemulsion [27], solid dispersion [21,28], co-spray dried composite powder formulation [29] and metal complexes [30] have been used to enhance its physicochemical properties of solubility and dissolution. The bitterness of AZT was masked by preparing microparticles [31] and nanoparticles without affecting dissolution rate [32].

Paracetamol (PCM, shown in Figure 1) exists in three polymorphic forms; a stable phase (I) and two metastable phases (II,III). Form I packs a herringbone motif, while forms II and III display layered structures [33,34]. It is used as an analgesic/antipyretic drug [35] and based on BCS it is classified class-III (low permeability, high solubility) API [36]. The crystal structure of PCM reveals hydrogen bonding, i.e., the crystal consists of potential H-bonding sites. Solid dosage forms of PCM may contain single-ingredient or in combination with other non-steroidal anti-inflammatory drugs (NSAIDs), opioids and antispasmodic drugs [37]. Although PCM contains no apparent basic or acidic sites, the existence of H-bond acceptor (oxygen in amide) and donor (OH and NH groups) makes them an excellent model drug for cocrystallization [38]. The cocrystallization of PCM has been extensively examined and several cocrystals have been successfully developed with coformers N,N-dimethyl-piperazine, piperazine, morpholine, 1,4-dioxane, 2,4-pyridinedicarboxylic acid and 4,4′-bipyridine [36].

A physician normally describes an antibiotic with an analgesic/antipyretic drug and the usual practice of simultaneous oral administration. Therefore, in the current study, an attempt has been made to synthesize cocrystal (AZT-PCM) system by solvent evaporation method. The possible intermolecular interactions and chemical compatibility of APIs was studied by Fourier transform infrared (FTIR), Raman spectroscopy and differential scanning calorimeter (DSC) techniques. New crystalline phase and thermal stability of the product were confirmed by powder X-ray diffraction (PXRD) and thermo gravimetric analysis (TGA), respectively. The molar ratio of components in the final product was confirmed by high-performance liquid chromatography (HPLC). Dissolution rate and antibacterial activity of cocrystals were studied and results were compared with raw azithromycin dihydrate (AZT DH) to evaluate its pharmaceutical applicability. The overall scheme of the study is given in Figure 2. 

## 2. Results and Discussion

### 2.1. Vibrational Spectroscopic Characterization

Cocrystal formation is the result of non-covalent interactions between different molecular components [39,40,41,42,43,44,45]. The existence of single component in the resulting bulk without structural variation is the key feature of the technique. It is therefore always observed that the vibrational (FTIR and Raman) spectroscopy is an excellent technique to characterize and study cocrystals.

#### 2.1.1. FTIR Spectroscopy

The FTIR analysis has been a valuable and extensively used tool for characterization and identification of new solid forms, in addition to other spectroscopic techniques [46]. The FTIR spectra (Figure 3) of raw drugs, physical mixture and cocrystal were obtained for possible interaction and chemical compatibility. The characteristic peaks pertaining to raw AZT DH can be assigned based on previously reported data [21,26,28]. The characteristic peaks observed at 3556 and 3486 cm^−1^ are assigned to OH-group, typically broad due to extensive intermolecular H-bonding, bands in the range 2750–3020 cm^−1^ indicate the presence of CH-aliphatic and 1718 cm^−1^ shows the presence of C=O stretching frequency. Signals at 1458 (CH_3_-O), 1373 (CH_2_-O), 1165 (C-O-C asymmetric stretch) and 1050 cm^−1^ (C-O-C symmetric stretch) are within the expected range. The characteristic peaks of PCM are observed at 3319 (OH), 3258 (NH), 3160–3032 (CH_3_), 1654 (C=O), 1610 (C=C), 1557 (NH bending), 1507 (CH asymmetric), 1443–1437 (C-C), 1368–1328 and 1260–1227 (symmetrical bending in CH and CN aryl stretching), 1171 and 965 cm^−1^ (C-O and C-N) which show agreement with the reported data [47,48].

Due to the complex FTIR spectrum of cocrystal (AZT-PCM) the unambiguous assignment of peaks is very tough task. However, attenuation, broadening and slight change in position of peaks can give a hint for Van der Waals interactions [49,50]. An abnormally broad peak formed in the region of 3600–3015 cm^−1^ indicates some new set of H-bond interactions [51,52], while no such peak formation is observed in the physical mixture spectrum. The OH of PCM is involved in intermolecular hydrogen bonding with N in pure solid state while in cocrystal the OH stretching peak (3319 cm^−1^) and NH bending peak (1557 cm^−1^) of PCM have shifted to 3327 and 1565 cm^−1^, respectively. The peak of AZT DH shifted from 1718 to 1727 cm^−1^ may be due to the involvement of C=O functional group in H-bonding, where slight shift of electrons density and deduction of bond order can be observed. From 1718–1458 cm^−1^, the AZT DH remains inactive [53] where several characteristic peaks that correspond to PCM appeared in this region [36]. An overall inspection of spectrum indicates the existence of PCM and shifting of OH and NH peaks revealed the establishment of hydrogen bonding with AZT. Moreover, slight shifting in wave number values of peaks was also observed in the fingerprint region. From the above discussion it is strongly suggested that AZT and PCM interact through non-covalent interactions with each other to afford cocrystals. Furthermore, both drugs showed compatibility with each other because all characteristic peaks of both components appeared in the FTIR spectrum of the resulted cocrystal (Figure 3).

#### 2.1.2. Raman Spectroscopy

The establishment of intermolecular interactions between cocrystal components leads to valuable changes in the Raman spectra, as shown in Figure 4. The Raman spectrum of AZT DH exhibited bands at positions 2961, 2930, 2880 and 2824 cm^−1^, while PCM had peaks at 3088, 3053 and 2919 cm^−1^. The peaks 2921 and 2930 cm^−1^ of AZT DH and PCM appeared at 2930 cm^−1^ (broad peak) and the doublet at 2955 and 2970 cm^−1^ of AZT DH appeared as a single peak with enhanced intensity at 2961 cm^−1^ in the spectrum of the corresponding cocrystal. The peaks at 3088 and 3055 cm^−1^ correspond to PCM almost disappeared in cocrystal while peaks of AZT DH at 2824 and 2880 cm^−1^ remained unchanged.

AZT DH has a peak at 1692 cm^−1^ corresponding to the C=O stretching, which shifted to a higher wave number 1712 cm^−1^ during cocrystallization. The same blue shift of carbonyl peak was also observed in the FTIR spectrum. The peak shifting indicates that the carbonyl group of AZT participates in hydrogen bonding. The amide (C=O) stretch and other characteristics peaks 1632, 1600, 1547 cm^−1^ of PCM appeared weak and broad at 1664, 1621, 1550 cm^−1^ in a cocrystal complex. Apart from these changes, several peaks of AZT DH and PCM in the stretching and bending vibration regions were shifted (frequency and intensity) in cocrystal spectrum, depending upon their participation in secondary interactions.

The obtained Raman spectral results are in strong agreement with previously published data for cocrystals [54,55]. Raman experimental results indicated that the crystalline phase of the cocrystal is not a simple combination of starting materials, but a different crystal phase due to hydrogen bonding interactions between AZT and PCM. Therefore, from Raman analysis it is further confirmed that AZT-PCM afford cocrystals as a result of non-covalent interactions.

### 2.2. PXRD Characterization

For detailed structural information, crystals with appropriate dimensions and size have to be prepared, which is often a hectic task and fails in many instances. Another limitation of single crystal analysis is that the selected crystal may represent a side product, not the desired polycrystalline product (bulk). Moreover, single crystal analysis is time consuming and not a readily available technique. Conversely, PXRD is a readily available technique generally used for confirmation and determination of bulk purity and crystallinity of the bulk material [56]. The DH form of AZT was re-crystalized from the same solvent system used in the preparation of cocrystals and subjected to PXRD analysis along PCM and cocrystals. Although the authentic data about cocrystals can be obtained from single crystal XRD diffractogram, in some cases where the crystal obtained is not suitable for single crystal XRD analysis, PXRD analysis is preferred to get the data about the crystals formed [57]. Referring to the study of Wu et al. [57], PXRD analysis has been used in this study to confirm the crystallinity of the product formed. Before the PXRD measurement, we crushed the single crystals formed to eliminate the effects of preferential orientation and enhance the clarity of the small diffraction peaks [57]. New solid phases were identified as changes were observed in the PXRD pattern, by comparing the starting material and final product [58]. The PXRD diffractograms of the prepared cocrystal and raw materials are given in Figure 5. The diffractograms of pure AZT and recrystallized AZT exhibited DH and MH forms, respectively which showed good agreement with previously reported experimental PXRD patterns [57]. The diffraction pattern of the cocrystal exhibited different pattern to that of starting materials and recrystallized AZT MH. The different PXRD patterns of cocrystals in comparison with the starting precursors indicate that the new crystalline phase material has been successfully formed with a different crystalline phase [57,59].

### 2.3. Thermal Analysis

#### 2.3.1. DSC Analysis

DSC analyses are extensively utilized for cocrystals confirmation. The thermal behavior of newly prepared cocrystals and corresponding raw materials were investigated using DSC (Figure 6). The endothermic peak of AZT DH and PCM were observed at 126.7 °C and 168.23 °C, respectively. Meanwhile, the endothermic peak at 143.27 °C along with shoulder peak provide enough insight regarding the formation of cocrystal [60]. Moreover, the shifting of endothermic peaks of AZT-PCM physical mixture (1:1, *w*/*w*) to lower temperature previously reported, which indicating strong interactions between APIs [61]. The survey of 50 samples was evaluated, the results showed 51% cocrystals melting point in between those of respective APIs and their coformer, while 39% were lower than both of the components, only for 4% the value was similar to either the API or conformer, and 6% had higher melting than the parent material. From the results it has been concluded that melting point of an API can shift due to the formation of cocrystals [62]. During the course of our study, the melting peak of the cocrystal was observed in between both components. Therefore, DSC analysis reveals the formation of new cocrystal material [63,64]. Furthermore, a different position of the melting peak also suggested physical interactions developed between both the APIs in their resulted cocrystal [61]. The binary phase diagram is also given in Appendix A shows the physical interaction between corresponding constituents. Additionally, the preliminary melting points were determined through melting point analyzer are given in Appendix A.

#### 2.3.2. Thermogravimetric Analysis (TGA)

The degree of hydration and thermal stability of cocrystal was evaluated by TGA. The obtained thermograms of the cocrystal and respective pure drugs are presented in Figure 7. The thermogram of the cocrystal exhibiting the transition of the cocrystal from a hydrated to anhydrous form. The observed weight loss (2.90%) up to onset temperature corresponds to the loss of water and other solvent residue molecules, as theoretically 1.96% and 3.84% water molecule weight loss is encountered while preparing AZT-PCM (1:1) MH and DH cocrystals, respectively. In case of semihydrate solvated form (AZT-PCM. H_2_O. ½CH_3_OH) [12], where methanol is also attached, the mass loss is theoretically 3.64%. The 2.90% mass loss up to onset temperature can be explained by the fact that when water/methanol mixture is used in formulation, the DH form of AZT is converted into MH form. MH form of AZT was previously obtained in a water/ethanol mixture [13] which is hygroscopic as mentioned earlier and the loss 2.90% may be due to some unbound water molecules along with bounded water molecules associated with MH form.

Previously compiled data revealed that AZT DH transformed to MH, followed by anhydrous form up to its melting temperature [65]. Similarly, the weight loss of cocrystal up to onset temperature attributed to the removal of unbound or bound (solvate) molecules from cocrystals [57]. Temperature above the melting point causes the degradation of cocrystals [66]. Therefore, the weight loss of AZT-PCM cocrystals before onset temperature may be due to the loss of water molecules from cocrystals or residual solvent elimination or water absorbed by the samples, while weight loss starting from 240 °C is due to degradation. Little weight loss was also observed after melting temperature and before degradation, which may be due the volatilization of components of the cocrystal.

### 2.4. HPLC Analysis

Molar ratio of olanzapine and carbamazepine in their nicotinamide cocrystals were determined by HPLC analysis [67]. Therefore, the molar ratio of AZT was determined in cocrystals using previously developed and validated HPLC method [68]. The AZT concentration in cocrystals was 81%, while theoretical concentration in case of cocrystal (combined in 1:1 ratio) is about 83%. Experimental and theoretical values are in good agreement therefore, it is suggested that cocrystals have successfully formed where both the synthons are present in equimolar ratios.

### 2.5. Powder Dissolution Study

The in vitro drug release studies of AZT DH and the prepared cocrystal were performed in phosphate buffer (pH 6.8), acetate buffer (pH 4.5) and 0.1 mM HCl. The result displayed in Figure 8 reveals that cocrystal exhibited a better dissolution rate than pure AZT DH. Due to agglomeration and poor wettability, pure AZT DH showed poor dissolution up to one hour [21]. However, the better dissolution performance of the resulting cocrystal can be traced to changes in size, shape, crystalline pattern [57] and hydrophilic nature of PCM [36].

### 2.6. Saturation Solubility Studies

The solubility of AZT DH in aqueous and phosphate buffer solution (pH 6.8) was calculated for pure drug and a cocrystal. The obtained results presented in Table 1 indicate that the solubility of AZT has significantly improved through cocrystallization.

### 2.7. Antibacterial Studies

The comparative zone of inhibition and MIC studies of the selected antibiotic and cocrystal were performed against *Escherichia coli* (*E. coli*), *Salmonella typhi* (*S. typhi*) and *Klebsiella pneumonia* (*K. pneumonia*) bacterial strains [43], the results obtained have been summarized in Table 2 and Table 3, respectively. Cocrystals exhibited better results than the parental drug even though a smaller amount of AZT exists in the cocrystal. It may be due to a faster dissolution rate [69] or due to AZT-PCM cocrystal physicochemical interactions [7] or increasing diffusion through the bacterial cell [70]. Moreover, PCM also exhibited mild antimicrobial effectiveness, which may contribute towards enhanced efficiency of antibacterial activity of cocrystals [71]. The activity of the cocrystal in terms of bacterial inhibition was improved; the technique is promising and can be modified for MDRS (multidrug resistant strains) in future studies. 

## 3. Experimental Section

### 3.1. Materials

AZT DH (C_38_H_72_N_2_O_12_.2H_2_O, 98%) and PCM (CH_3_CONHC_6_H_4_OH, 98%) were obtained from the local pharmaceutical industry. Methanol (HPLC grade), HCl (37%) and KH_2_PO_4_ were purchased from Sigma Aldrich, while filtered, double distilled water was purchased from a local market. All chemicals were used without a purity test or further processing.

### 3.2. Synthesis of Cocrystals

Solution of AZT DH and PCM in water/methanol (1:1, *v*/*v*) solvent system was prepared and sonicated for 10 min at 40 °C; it was allowed to cool to room temperature in the dark. Crystalline materials obtained after a few days were separated from the solution and were dried under the vacuum at 40 °C. The prepared cocrystals were stored in the dark (as a precautionary measure) for further analysis.

### 3.3. Preparation of Physical Mixture

The physical mixture of AZT DH and PCM was prepared in equimolar amounts. Both drugs were transferred into micro tube and mixed thoroughly (for better homogeneity) for 15 min at 30 rpm using vortex mixer. The sample was stored at ambient temperature till further studies.

### 3.4. Vibrational Studies

FTIR analyses were performed using FTIR spectrophotometer (PerkinElmer spectrum-10.5.1). The FTIR spectra of AZT DH, PCM, cocrystals and physical mixture were recorded in the range of 4000–400 cm^−1^. Raman spectra were obtained using Lab RAM HR, Horiba JobinYvon, France. The spectra of pure drugs and cocrystals were recorded in the range of 100–3200 cm^−1^.

### 3.5. Powder X-ray Diffraction (PXRD)

The PXRD analysis for cocrystals and starting precursors were performed using JDX 3532 Jeol Japan X-Ray diffractometer. The samples were scanned at 2θ from 6–40° with 0.05° increments and counts were accumulated for 1 s per step.

### 3.6. Thermal Analysis

Thermal responses of cocrystals and respective drugs were measured by DSC-60 (Shimadzu, Japan) taking 4.6 mg of the sample. Under constant nitrogen gas flow, the samples were heated at the rate of 10 °C/min in the range of 25 to 250 °C. The STARe software was used for data processing and analysis.

Thermal gravimetric study was done with the help of Diamond Series TG/DTA Perkin Elmer, USA analyzer using Al_2_O_3_ as reference. The sample was heated at a rate of 10 °C/min in the range of 40–300 °C.

### 3.7. HPLC Analysis

The concentration of AZT was measured in its cocrystals using a previously reported method [67]. The analysis was performed using Agilent-1260 instrument with the PDA (Shimadzu, Japan). The method was performed on reversed phase column ZORABEX Eclipse Plus C18 (4.6 × 250 mm, 5 µm). Methanol:phosphate buffer (9:1, *v*/*v*) was used as a mobile phase at a flow rate of 1.2 mL/min with a PDA detector, 210 nm. The temperature was adjusted to 40 °C with an injection volume of 50 µL. The AZT DH (1 mg/mL) and an equivalent concentration of the respective cocrystal solution were prepared in the mobile phase. Both solutions were run, and the concentration of AZT in cocrystals was measured.

### 3.8. In Vitro Powder Dissolution

Dissolution tests of AZT DH in pure state and cocrystal were carried out using USP apparatus type-2 (paddle method). An ADV 06 was operated under the sink condition. Cocrystals and drug were converted to fine powder and sieved through 100-mesh sieves to reduce effect of size on dissolution rate. An exact amount, 100 mg of the drug and an equivalent amount of the cocrystal was taken and added to the vessels of dissolution apparatus containing 900 mL phosphate buffer pH 6.8. The rotation speed of the paddle was adjusted to 75 rpm at 37 °C and aliquots of 5 mL, withdrawn at predetermine time interval up to 60 min. After withdrawing aliquots, the same volume of dissolution medium was added to vessel. The same procedure was followed for acetate buffer (pH 4.5) and 0.1 mM HCl dissolution medium. The samples were filtered, and percent drug releases were analyzed by HPLC. All the dissolution tests were performed in triplicate, and mean values were noted.

### 3.9. Saturation Solubility Studies

To study saturation solubility of AZT DH and cocrystal, saturation solubility test was carried out. Excess amount of the drug and the cocrystal were added to 100 mL of distilled water and a phosphate buffer solution (pH 6.8). Samples were stirred at 100 rpm at 37 °C for 24 h, filtered, suitably diluted and analyzed by HPLC. The saturation solubility determination experiments were performed in triplicate and respective mean values were noted.

### 3.10. In Vitro Antibacterial Study

Petri dishes and an agar solution were sterilized in autoclave. A known volume of 20 mL agar solution under sterile conditions was added to each petri dish and incubated for 24 h. The plates with no contamination were selected for further study. The bacterial strains (*E. coli*; MTCC 1687, *S. typhi*; MTCC 734, *K. pneumonia*; MTCC 1030) were inoculated to plates with the help of sterile swabs. The drug and equivalent amount of cocrystals (1000 µg/mL) solutions were separately prepared and further diluted to different concentrations (500, 250 and 125 µg/mL). From each solution 5 µL were added to 6 mm filter paper discs. After ten minutes of drying, drug loaded (5, 2.5, 1.25, 0.62 µg) filter paper discs were placed on agar plates surface using sterile forceps. The plates were incubated (37 °C) for 24 h and diameters indicating the zone of inhibition were manually noted. The antibacterial activity was performed in triplicate and mean zone of inhibitions were determined.

The minimal inhibitory concentration (MIC) of the prepared cocrystal and pure AZT were also determined against *E. coli*, *S. typhi* and *K. pneumonia* using the broth dilution method. Two-fold serial dilutions of AZT DH and cocrystal were made in sterile nutrient broth to give concentrations ranging from 2–256 μg/mL. Ten sterile tubes were placed on a rack, each labeled 1 through 8, along with a positive control and negative control. About 1 mL of different concentrations of the drug were added in to separate test tubes and inoculated with 1 mL bacterial suspensions (10^6^ CFU/mL) of selected strain. The inoculated tubes were incubated at 37 °C for 24 h after which they were inspected for turbidity. A positive control (growth) was formed by culture broth with microorganisms while the negative control (sterility) consisted of broth with no microorganisms. 

The micrometric analysis details are elaborated in Appendix A with summarized results in Appendix A, giving useful information about resultant products.

## 4. Conclusions and Future Work

Cocrystallization for designing and synthesis of DDCs is of pivotal and immense importance in pharmaceutical industry to manipulate physicochemical properties of APIs. In this work, AZT-PCM cocrystals were prepared and successfully isolated using solvent evaporation method. The characterization techniques such as FTIR, Raman, PXRD, DSC, TG and HPLC were successfully employed for cocrystal confirmation. Cocrystal exhibited better dissolution, solubility and antimicrobial performance than parental AZT DH drug. However, the effectiveness of the synthesized cocrystal needs in vivo confirmation. Though cocrystals were preliminary confirmed by different characterization techniques, for detailed structural characterization and discussion on exact interaction and related information, single XRD analysis is warranted. The field of drug-drug cocrystallization has bright chances to emerge in future studies. The efficiency was improved against the bacterial strains. The heterosynthons is efficient against *K. pneumonia* followed by *E. coli* and *S. typhi*.

## Figures and Tables

**Figure 1 antibiotics-10-00939-f001:**
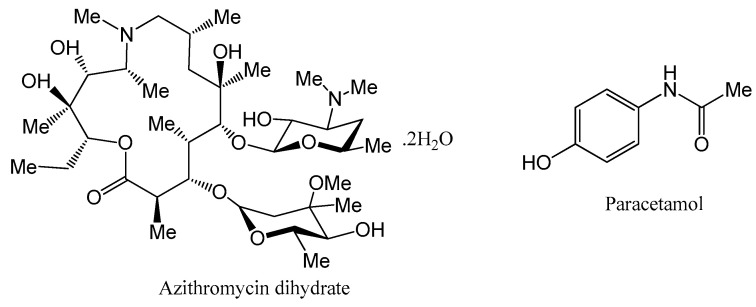
Chemical structures of azithromycin dihydrate (AZT DH) and paracetamol (PCM).

**Figure 2 antibiotics-10-00939-f002:**
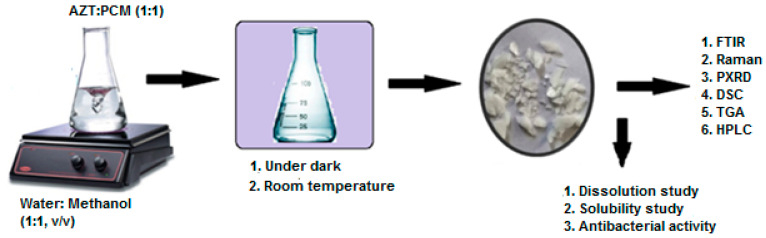
Scheme of study for synthesis, characterization and in vitro evaluation of cocrystal.

**Figure 3 antibiotics-10-00939-f003:**
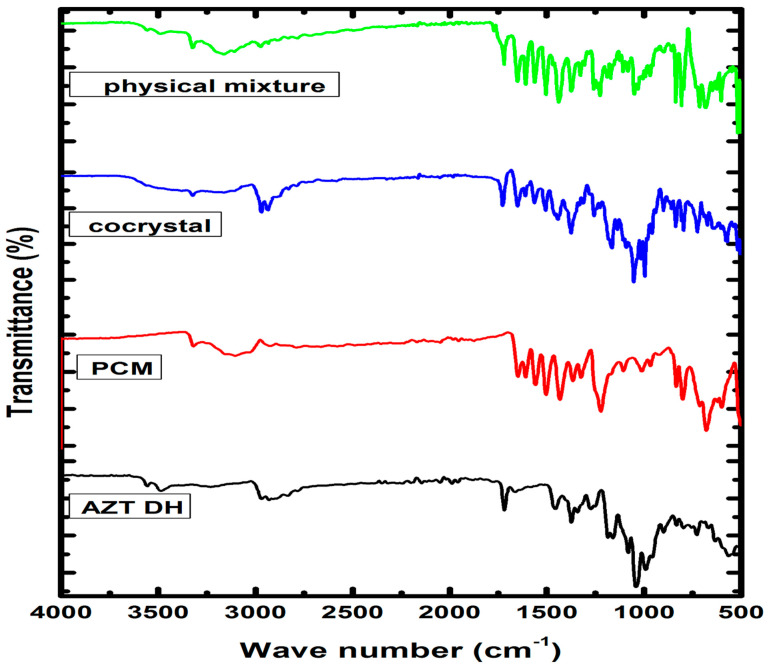
FTIR spectra of AZT DH, PCM, cocrystal and physical mixture.

**Figure 4 antibiotics-10-00939-f004:**
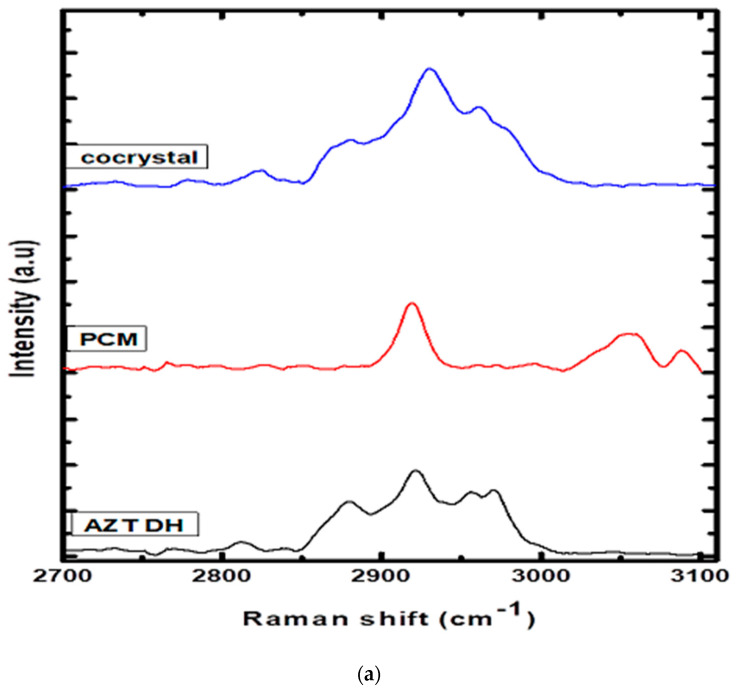
(**a**) Raman spectra of AZT DH, PCM and cocrystal range (2700–3100 cm^−1^). (**b**) Raman spectra of PCM, AZT DH and the cocrystal range (500–1750 cm^−1^).

**Figure 5 antibiotics-10-00939-f005:**
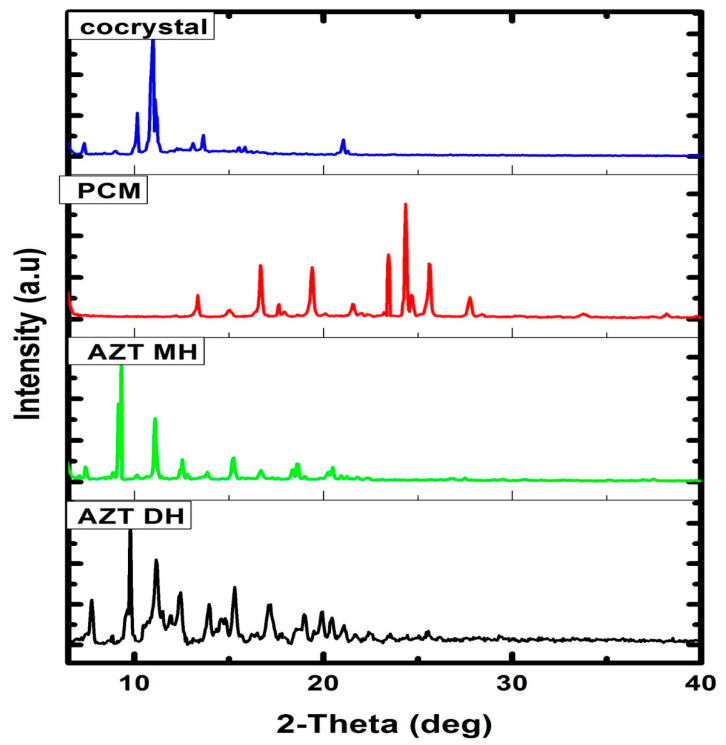
PXRD patterns of AZT DH, AZT MH, PCM and cocrystal.

**Figure 6 antibiotics-10-00939-f006:**
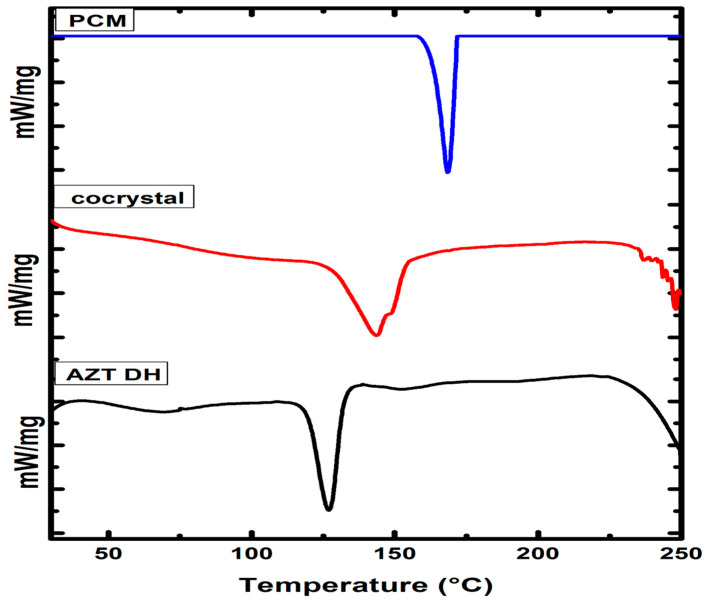
The DSC thermograms AZT DH, PCM and cocrystal.

**Figure 7 antibiotics-10-00939-f007:**
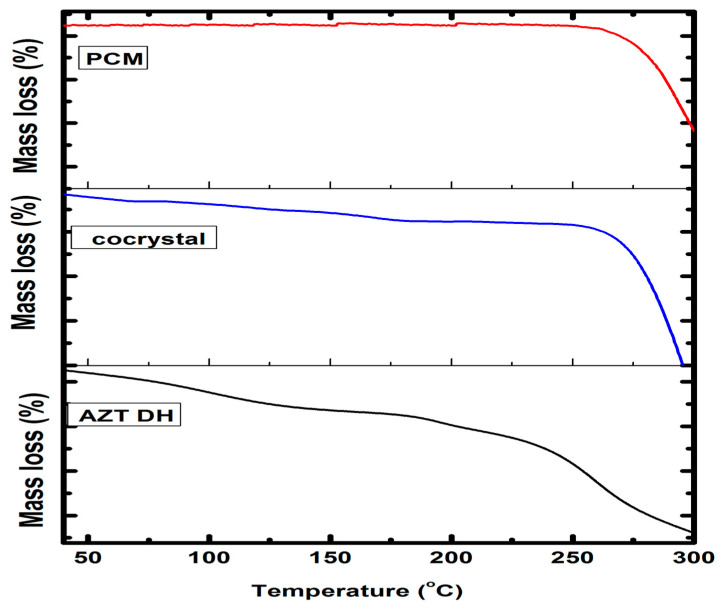
TGA thermograms of AZT DH, PCM and cocrystal.

**Figure 8 antibiotics-10-00939-f008:**
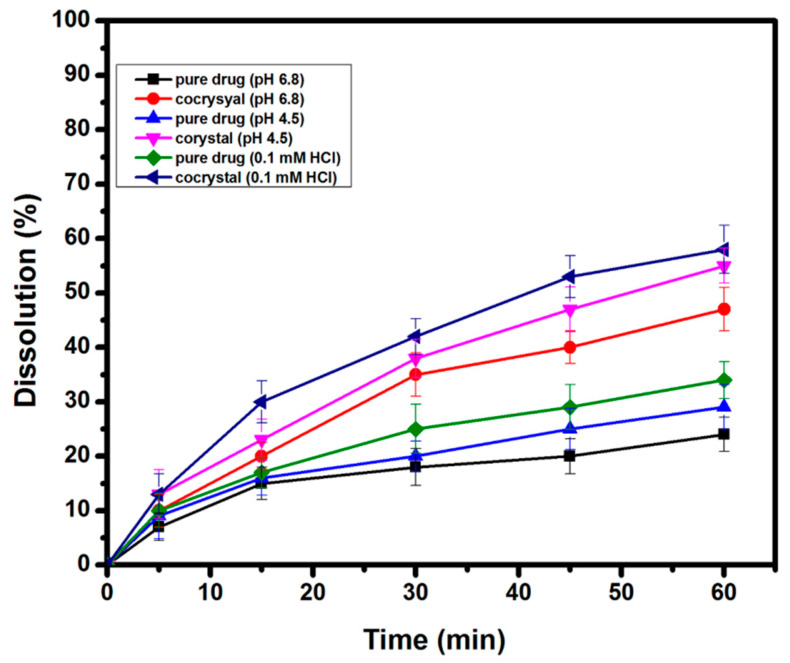
Powder dissolution rate of AZT DH and cocrystal in different dissolution medium.

**Table 1 antibiotics-10-00939-t001:** Saturation solubility of AZT DH and cocrystal.

Solubility Medium	Pure Drug (μg/mL)	Cocrystal (μg/mL)
Distilled water	55 ± 1.31	7 ± 1.42 ***
Phosphate Buffer (pH 6.8)	60 ± 1.28	80 ± 1.40 ***

*** indicates significant difference (*p* < 0.05) when t-test is applied for comparing the solubility values of cocrystal with that of a pure drug.

**Table 2 antibiotics-10-00939-t002:** Antimicrobial activity of AZT DH and cocrystal against different bacterial strains.

Bacterial Strain	Sample Amount (µg)	AZT DH (100%) Zone of Inhibition (mm)	Cocrystal (83%) Zone of Inhibition (mm)
*K. pneumonia*	5	21 ± 0.45	23 ± 0.41 **
	2.5	17 ± 0.40	18 ± 0.37 *
	1.25	14 ± 0.35	15 ± 0.30 **
	0.62	10 ± 0.25	12 ± 0.23 ***
*E. coli*	5	15 ± 0.32	16 ± 0.27 *
	2.5	10 ± 0.30	12 ± 0.25 ***
	1.25	8 ± 0.25	9 ± 0.27 **
	0.62	--	--
*S. typhi*	*5*	13 ± 0.35	13 ± 0.31 *
	*2.5*	9 ± 0.30	10 ± 0.27 **
	*1.25*	7 ± 0.24	7 ± 0.21 *
	*0.62*	--	--

*** *p* ≤ 0.0005, ** *p* ≤ 0.005, while * *p* ≤ 0.05. Percent concentration of active antibiotic in the formulations presented in the brackets in the first row of columns two and three of the table.

**Table 3 antibiotics-10-00939-t003:** MIC values of pure AZT DH and cocrystal against different bacterial strains.

Sample	MIC (μg/mL)		
	*E. coli*	*S. typhi*	*K. pneumonia*
AZT DH (100%)	64	128	64
Cocrystal (83%)	64	64	32

In the brackets in the first column of the table are the percent concentrations of active antibiotic in the formulations.

## Data Availability

All the data associated with this paper has been presented in this manuscript. Not data in any repository is available to be linked here.

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
