# Peer review of "Enhancing Dissolution Rate and Antibacterial Efficiency of Azithromycin through Drug-Drug Cocrystals with Paracetamol"

_antibiotics, 2021, doi:10.3390/antibiotics10080939_

Round 1

Reviewer 1 Report

The authors reports about the synthesis and characterization of a new drug-drug cocrystal made of azithromycin and paracetamol in 1:1 ratio.

Even though cocrystals are largely reported in the pharmaceutical arena, there is no example of Azithromycin cocrystals. However, because the background in the pharma area is quite large, I strongly encourage the authors to improve the citation by adding the references reported in the pdf file attached.

Even though the experimental part is fully supported it is a pity not to see the crystal structure of the cocrystal. I know it might be hard to get a single crystal however, they could try to solve the structure from XRPD. 

Finally, I would suggest the publication of this paper after minor correction as reported in the pdf.

Author Response

Reviewer 1:

The authors report about the synthesis and characterization of a new drug-drug cocrystal made of azithromycin and paracetamol in 1:1 ratio.Even though cocrystals are largely reported in the pharmaceutical arena, there is no example of Azithromycin cocrystals. However, because the background in the pharma area is quite large, I strongly encourage the authors to improve the citation by adding the references reported in the pdf file attached.

  • All the citations suggested by the worthy review are cited at appropriate places and highlighted in the list of references.

Even though the experimental part is fully supported it is a pity not to see the crystal structure of the cocrystal. I know it might be hard to get a single crystal however, they could try to solve the structure from XRPD.

  • Yes, We know the fact that single crystal is extremely important, but our repeated attempts were in vain. It is felt that enough data have been collected to justify the claim. Necessary amendments have been carried out (highlighted) to avoid ambiguities as suggested. Further all type and grammatical corrections were gladly incorporated.
  • Worthy reviewer, for single crystal XRD the sample was send to Korea, Oman and Germany. Unfortunately, due to present pandemic we could not succeeded.

Finally, I would suggest the publication of this paper after minor correction as reported in the pdf

We thank you for your efforts and time in improving the Draft. All changes suggested in the PDF file were incorporated accordingly in the revised manuscript. 

Reviewer 2 Report

The present manuscript deals with the preparation of a drug-drug cocrystal between azithromycin and paracetamol, characterization and powder dissolution and antibacterial studies.

I recommend to reject the manuscript in Antibiotics as the document requires an overall revision through all its sections and the formation of the cocrystal is not confirmed by the results described.

First, both starting products, PCM or AZT show different solid forms (anhydrous and solvated forms including hydrates). However, no comments are included in the document. A full revision should be included.

In the case of AZT, the initial solid form used is described in the experimental section. Which polymorph is used in the case of paracetamol? Please include it. The authors do not describe how the physical mixture was prepared.

In Figure 5, the XRD pattern of AZT dihydrate (this is the form used according to the Experimental section) does not match with the simulated PXRD from single crystal structure for this hydrate (for instance, refcode GEGJAD). Were the samples ground? Did authors consider particle size, textural effects,..?

The pattern obtained for the mentioned cocrystal should also be compared with all the other forms described for the precursors not only the forms used in order to discard that the resulting PXRD does not contain a mixture of forms. Which are the new peaks observed in the XRD? Most of the peaks observed match with the ones shown in the AZT or PCM patterns included.

Solvent evaporation is a good method among others for single crystal preparation and its structure resolution. Single crystal XRD of the cocrystal is highly recommended (or even mandatory in this case). The simulated powder pattern for the solved crystal structure could be compared and this would confirm that the bulk solid and the single crystal are the same compound. Specially taking into account that in the case of AZT several solvates are described and only it is possible to distinguish them by this technique as their powder patterns are identical (see references Z. Kristallog. 2003, 218, 703; Z. Zristallog. 2005, 220, 66 or Z. Kristallog. 2007, 222, 492).

The authors should revise carefully all the FTIR assignments and the literature to correct errors. For instance, in the case of PCM the bands of OH and NH are exchanged.

Regarding thermal analysis, some information is missing. Authors do not give the melting peaks (or onset values) for their samples but the ones describe in the literature, not even for the as-prepared cocrystal. And in the TGA, why the loss is assigned to water and not to methanol or a mixture of both solvents?  Moreover, the value obtain does not match with a monohydrate or a dihydrate. The authors should confirm the water content by another technique, as Karl-fisher method.

For sections 2.5, 2.6 and 2.7, have replicates been performed for those techniques? please include it.

Are differences among antimicrobial activity values shown in Table 2 significant? For instance, for E. coli and 5 micrograms, AZT, 15 mm and for cocrystal, 16 mm.

Finally, the document should carefully revise as it contains many typing errors. A few examples: author affiliation number 3 is not assigned to any author, in figure 1 or reference 33 authors are not included,..

Author Response

Reviewer 2:

The present manuscript deals with the preparation of a drug-drug cocrystal between azithromycin and paracetamol, characterization and powder dissolution and antibacterial studies.

I recommend to reject the manuscript in Antibiotics as the document requires an overall revision through all its sections and the formation of the cocrystal is not confirmed by the results described.

  • Worthy reviewer, we have revised the manuscript according to the provided suggestion of the reviewers. Hopefully, it will be acceptable now.

First, both starting products, PCM or AZT show different solid forms (anhydrous and solvated forms including hydrates). However, no comments are included in the document. A full revision should be included.

  • The AZT different solid forms (hydrated and solvated) [1-5] and PCM polymorphs [7,6] have been discussed and relevant literature is added.
  1. Montejo-Bernardo, J.M.; García-Granda, S.; Bayod-Jasanada, M.S.; Llavona-Díaz, L.; Llorente, I. On the Solid State Conformation of Azithromycin Monohydrate and Dihydrate Pseudopolymorphs. Zeitschrift fur Krist.2005, 220, doi:10.1524/zkri.220.1.66.58890.
  2. Neglur, R.; Hosten, E.; Aucamp, M.; Liebenberg, W.; Grooff, D. Water and the Relationship to the Crystal Structure Stability of Azithromycin: Thermal Investigations of Solvatomorphism, Amorphism and Polymorphism. Therm. Anal. Calorim.2018, 132, doi:10.1007/s10973-017-6928-6.
  3. Montejo-Bernardo, J.M.; García-Granda, S. New Solvated Form of the Antibiotic Azithromycin. Clues about the Role of the Water Retained inside the Crystal. Zeitschrift fur Krist.2007, 222, doi:10.1524/zkri.2007.222.9.492.
  4. Montejo-Bernardo, J.M.; García-Granda, S.; Bayod-Jasanada, M.S.; Llavona-Díaz, L.; Llorente, I. X-Ray Study of the Pseudopolymorphism of the Azithromycin Monohydrate. Zeitschrift fur Krist.2003, 218, doi:10.1524/zkri.218.10.703.20761.
  5. Blanco, M.; Valdés, D.; Llorente, I.; Bayod, M. Application of NIR Spectroscopy in Polymorphic Analysis: Study of Pseudo-Polymorphs Stability. Pharm. Sci. 2005, 94, doi:10.1002/jps.20362.
  6. Nicnols, G.; Frampton, C.S. Physicochemical Characterization of the Orthorhombic Polymorph of Paracetamol Crystallized from Solution. Pharm. Sci.1998, 87, doi:10.1021/js970483d.
  7. Perrin, M.A.; Neumann, M.A.; Elmaleh, H.; Zaske, L. Crystal Structure Determination of the Elusive Paracetamol Form III. Commun.2009, doi:10.1039/b822882e.

In the case of AZT, the initial solid form used is described in the experimental section. Which polymorph is used in the case of paracetamol? Please include it. The authors do not describe how the physical mixture was prepared.

  • Azithromycin dihydrate (AZT DH) and Paracetamol (monoclinic form-1 which is stable and commercially available) were used during this study. The AZT DH was mistakenly written azithromycin dehydrate in experimental section which is now corrected. Physical mixture preparation procedure was added to experimental section.

In Figure 5, the XRD pattern of AZT dihydrate (this is the form used according to the Experimental section) does not match with the simulated PXRD from single crystal structure for this hydrate (for instance, refcode GEGJAD). Were the samples ground? Did authors consider particle size, textural effects?

  • We obtained the PXRD diffractogram of micronized AZT DH, which probably results some important peaks disappearance. However, the unprocessed AZT DH diffractogram was revised and Figure-5 is now updated. The characteristic peaks of AZT.DH (5-30°, 2-theta) show good agreement with experimental reported data (references given below and added to the list in the manuscript).
  1. Wu, S.; Shen, H.; Li, K.; Yu, B.; Xu, S.; Chen, M.; Gong, J.; Hou, B.H. Agglomeration Mechanism of Azithromycin Dihydrate in Acetone-Water Mixtures and Optimization of the Powder Properties. Ind. Eng. Chem. Res.2016, 55, doi:10.1021/acs.iecr.5b04437.
  2. Adeli, E. Preparation and Evaluation of Azithromycin Binary Solid Dispersions Using Various Polyethylene Glycols for the Improvement of the Drug Solubility and Dissolution Rate. Brazilian J. Pharm. Sci.2016, 52, doi:10.1590/S1984-82502016000100002.
  3. Timoumi, S.; Mangin, D.; Peczalski, R.; Zagrouba, F.; Andrieu, J. Stability and Thermophysical Properties of Azithromycin Dihydrate. Arab. J. Chem.2014, 7, doi:10.1016/j.arabjc.2010.10.024.

The pattern obtained for the mentioned cocrystal should also be compared with all the other forms described for the precursors not only the forms used in order to discard that the resulting PXRD does not contain a mixture of forms. Which are the new peaks observed in the XRD? Most of the peaks observed match with the ones shown in the AZT or PCM patterns included.

  • Two new peaks were observed at 36.12° and 46.06° which we believe to belong to the new product. The obtained PXRD of cocrystal showed different pattern from the AZT (solvates exhibited almost similar PXRD pattern) and PCM polymorphs. This indicates the formation of new phase of material which is different from all forms of starting material.

Solvent evaporation is a good method among others for single crystal preparation and its structure resolution. Single crystal XRD of the cocrystal is highly recommended (or even mandatory in this case). The simulated powder pattern for the solved crystal structure could be compared and this would confirm that the bulk solid and the single crystal are the same compound. Specially taking into account that in the case of AZT several solvates are described and only it is possible to distinguish them by this technique as their powder patterns are identical (see references Z. Kristallog. 2003, 218, 703; Z. Zristallog. 2005, 220, 66 or Z. Kristallog. 2007, 222, 492).

  • Worthy reviewer, for single crystal XRD the sample was send to Korea, Oman and Germany. Unfortunately, due to present pandemic we could not succeeded.
  • Yes, We totally agree that SC XRD is the ultimate solution in many cases to address the questions being raised by the reviewer. At this stage we request our reviewer to relax the condition, which is beyond our control and we are helpless under these conditions. Our work for other systems is in progress and crystals have been obtained. The results will be published in near future once the data set is complete.

The authors should revise carefully all the FTIR assignments and the literature to correct errors. For instance, in the case of PCM the bands of OH and NH are exchanged.

  • We based the assignment of OH and NH on the bases of literature reports pertaining to the related work[1, 2]. The peaks of OH and NH are exchanged accordingly.
  1. Srivastava, K.; Shimpi, M.R.; Srivastava, A.; Tandon, P.; Sinha, K.; Velaga, S.P. Vibrational Analysis and Chemical Activity of Paracetamol-Oxalic Acid Cocrystal Based on Monomer and Dimer Calculations: DFT and AIM Approach. RSC Adv.2016, 6, doi:10.1039/c5ra24402a.
  2. Trivedi, M. K., Patil, S., Shettigar, H., Bairwa, K., & Jana, S. Effect of Biofield Treatment on Spectral Properties of Paracetamol and Piroxicam. Chem. Sci. J.2015, 6, doi:10.4172/2150-3494.100098.

Regarding thermal analysis, some information is missing. Authors do not give the melting peaks (or onset values) for their samples but the ones describe in the literature, not even for the as-prepared cocrystal. And in the TGA, why the loss is assigned to water and not to methanol or a mixture of both solvents?  Moreover, the value obtain does not match with a monohydrate or a dihydrate. The authors should confirm the water content by another technique, as Karl-fisher method.

  1. The relevant information have been inserted accordingly in the respective section as: The observed weight loss (2.90%) up to onset temperature corresponds to loss of water and other solvent residue molecules as theoretically 1.96 and 3.84% weight losses of water molecules are encountered while preparing AZT-PCM (1:1) monohydrate and dihydrate cocrystals, respectively. In case of semihydrate solvated form where methanol is also attached the mass loss theoretically 3.64%. The 2.90% mass losses up to onset temperature, can be explained by fact that when water methanol mixture is used in formulation the dihydrate form of AZT is converted into monohydrate form [13] which is hygroscopic as mentioned earlier and the loss 2.90% may be due to some unbound water molecules along with bounded water molecules associated with monohydrate form.

For sections 2.5, 2.6 and 2.7, have replicates been performed for those techniques? please include it.

  • Corrected accordingly

Are differences among antimicrobial activity values shown in Table 2 significant? For instance, for E. coli and 5 micrograms, AZT, 15 mm and for cocrystal, 16 mm.

  • Worthy reviewer, there are no significant differences in terms of obtained results however, despite low concentration of antibiotics (paracetamol contributed about 17% mass in cocrystal) in cocrystal samples than pure antibiotics, the antibacterial performances of cocrystal is considerably improved and significant differences in this term are then high.

Finally, the document should carefully revise as it contains many typing errors. A few examples: author affiliation number 3 is not assigned to any author, in figure 1 or reference 33 authors are not included,

  • The manuscript was carefully revised and several minor missing were corrected.

At the end we are thankful to extremely constructive criticism and hope that the reviewer will consider our revision.

  • Thank you worthy reviewer, we have revised the manuscript according to your worthy suggestion.

Reviewer 3 Report

Some minor style changes are required as shown in the attached pdf file of the manuscript. In particular, please separate quantities from units as indicated.

Please check the references because in some articles the information is not complete. 

Author Response

Some minor style changes are required as shown in the attached pdf file of the manuscript. In particular, please separate quantities from units as indicated. Please check the references because in some articles the information is not complete. 

  • All the minor suggestion of the reviewer were incorporated with many thanks for their efforts.
  • Thank you worthy reviewer, the entire list has been checked and corrected accordingly.

I noticed the authors used a crude zone of inhibition assay to compare the antimicrobial effectiveness of two different formulations. I would consider this to be a crude measure and the difference observed are too small to make judgements re superiority. I would suggest that a more accurate method is used to measure actual MIC eg. broth micro dilution.

  • Worthy reviewer, the MIC was determined accordingly and results have been incorporated in table 3.
  • We are thankful to anonymous reviewer for their sincere efforts, precious time and constructive criticism. They helped us a lot to improve this piece of our work. We work in laboratories with very limited resources and equipment which hinder to compile a study with due perfection. Moreover, all possible efforts have been made to satisfy them and it is hoped that they will understand our limitations. It is clarified that we do not want and request them to compromise the scientific and professional quality.

Round 2

Reviewer 2 Report

Despite the efforts performed by the authors to improve the manuscript there are still some key points unclear. I should insist on the importance to determine that a new cocrystal is obtained before its used for powder dissolution or antibacterial studies.

The first and most important is the characterization by PXRD, which should be the fingerprint of the cocrystal. The diffractogram supplied only shows that there are two new peaks at high angles (36.12 and 46.06º) but not that a pure new phase is obtained containing the cocrystal. The powder pattern could correspond to a mixture of phases (cocrystal + precursors, precursors in a different phase (solvated forms) than the former used,). It has not been proved that no other phases are included. Was the solid obtained by evaporation ground in a mortar or micronized? Did the authors observed the solid obtained by evaporation by microscope? This could afford them a lot of information to detect different habits, which could be due to different solid forms. Strong efforts to obtain single crystals are recommended.

In the case of Figure 6, the DSC trace observed for the cocrystal is not a single peak but there are two overlapped peaks or a peak with a shoulder, so a more complex event. Comments included in the TGA section does not confirm cocrystal formation “before onset temperature may be due to the loss of water molecule from cocrystal or residual solvent elimination or water absorbed by the samples, while loss of mass after melting is due to degradation”. In the DSC there is no degradation till 225ºC, however in the TGA some loss in still observed before this temperature and strong degradation is observed around 250ºC. Could the authors clarify the following comment:  ‘In case of semihydrate solvated form where methanol is also attached the mass loss theoretically 3.64%.’  how is obtained this value?

Was the physical mixture analysed by TGA and DSC? How different were the thermograms of the physical mixture in comparison to the ones of cocrystal?

Preparation of physical mixture is described as follows: “… equimolar amount of both drugs were mixed well (for better homogeneity the mixture) and resulting solid mass was then crushed gently…”.  This description is unclear. The mixing was in a vortex, a drum mixer, in a mortar. Why was the sample dried? Or performed by triplicate?

Finally, although the links to DOI number are included, none of the references in the paper contain the page ranges as requested in the Instructions for authors. Please amend them.

New authors are included. Or Acknowledgements have also been modified.

Some typing or grammatical errors must be corrected (a physician describes instead of prescribes, azithromycin in figure 1, room temperature in figure 2 and 7, Vander Waals,. Unfinished sentence at the end of section 2.3.1, errors in the legend in Figure 8,.. Please revise carefully the document.

Author Response

Despite the efforts performed by the authors to improve the manuscript there are still some key points unclear. I should insist on the importance to determine that a new cocrystal is obtained before its used for powder dissolution or antibacterial studies.

The first and most important is the characterization by PXRD, which should be the fingerprint of the cocrystal. The diffractogram supplied only shows that there are two new peaks at high angles (36.12 and 46.06º) but not that a pure new phase is obtained containing the cocrystal. The powder pattern could correspond to a mixture of phases (cocrystal + precursors, precursors in a different phase (solvated forms) than the former used,). It has not been proved that no other phases are included. Was the solid obtained by evaporation ground in a mortar or micronized? Did the authors observed the solid obtained by evaporation by microscope? This could afford them a lot of information to detect different habits, which could be due to different solid forms. Strong efforts to obtain single crystals are recommended.

  • If mixture of components (cocrystal + precursors are polymorphs) present than the DSC thermogram should be multiple peaks while in our case it doesn’t happen. We obtained the material apparently in crystalline form through slow evaporation, the crystalline sample was selected for analysis.
  • We would repeatedly say that so far out attempts did not prove fruitful in obtaining good quality crystals for structural analysis, still we try to get but can not say when we will succeed.
  • The following literature in line with our results report some of the examples, if the reviewer agrees we are willing to incorporate some or all of them in our paper.
  • The FBX-PXM system exhibited distinctive Powder X-ray Diffraction (PXRD) pattern compared to that of FBX, PXM and their physical mixture (FBX-PXM PM) further confirming the formation of cocrystal.1 In the PXRD patterns of both solid forms, (±)DMY-caffeine and (±)DMY-urea, characteristic peaks of the starting materials are absent while new peaks appeared.2 The diffractograms of this system differ significantly from those obtained from its precursors. The CIP–PZCA systems synthesized at 15 Hz (NG and LAG) and the system prepared at 30 Hz (NG) show a diffraction pattern very similar to each other, although the latter showed amorphization during the mechanochemical synthesis , as suggested by the diffraction halo in its diffractogram. These samples have new diffraction peaks in 2θ equal to 5.5°, 18.9°, 26.4°, indicating the presence of a new crystalline phase, which is an evidence of the cocrystal formation.3

  1. Modani, S. et al. Generation and Evaluation of Pharmacologically Relevant Drug-Drug Cocrystal for Gout Therapy. Cryst. Growth Des. 20, (2020).
  2. Wang, C. et al. Enhancing Bioavailability of Dihydromyricetin through Inhibiting Precipitation of Soluble Cocrystals by a Crystallization Inhibitor. Cryst. Growth Des. 16, (2016).
  3. de Almeida, A. C. et al. Mechanochemical synthesis, characterization and thermal study of new cocrystals of ciprofloxacin with pyrazinoic acid and p-aminobenzoic acid. J. Therm. Anal. Calorim. 140, (2020).

  • Co-crystal formation in many drugs was previously reported based on evidence from the difference in the X-ray diffraction pattern.
  1. Wang L, Tan B, Zhang H, Deng Z. Pharmaceutical Cocrystals of Diflunisal with

Nicotinamide or Isonicotinamide. Org Process Res Dev 2013;17:1413−1418.

  1. Arafa MF, El-Gizawy SA, Osman MA, El Maghraby GM. Sucralose as co-crystal

co-former for hydrochlorothiazide: Development of oral disintegrating tablets. Drug Dev Ind Pharm 2016. doi 10.3109/03639045.2015.1118495.

  1. El-Gizawy, SA, Osman MA, Arafa MF, El Maghraby GM. Aerosil as a novel cocrystal

co-former for improving the dissolution rate of hydrochlorothiazide. Int J Pharm 2015;478:773–778.

  1. Hickey MB, Peterson ML, Scoppettuolo LA, Morrisette SL, Vetter A, Guzmán H,

Remenar JF, Zhang Z, Tawa MD, Haley S, Zaworotko MJ, Almarsson Ö. Performance comparison of a co-crystal of carbamazepine with marketed product. Eur J Pharm Biopharm 2007;67:112–119.

In the case of Figure 6, the DSC trace observed for the cocrystal is not a single peak but there are two overlapped peaks or a peak with a shoulder, so a more complex event. Comments included in the TGA section does not confirm cocrystal formation “before onset temperature may be due to the loss of water molecule from cocrystal or residual solvent elimination or water absorbed by the samples, while loss of mass after melting is due to degradation”. In the DSC there is no degradation untill 225ºC, however in the TGA loss is observed before this temperature and strong degradation is observed around 250ºC. Could the authors clarify the following comment:  ‘In case of semihydrate solvated form where methanol is also attached, the mass loss theoretically 3.64%.’  how is obtained this value?

  • Though most of the cases the DSC thermogram of cocrystal exhibiting single endothermic peak. However, several (two or three) endothermic peaks of cocrystal has also been reported in literature.
  1. Nicolov, M.; Ghiulai, R.M.; Voicu, M.; Mioc, M.; Duse, A.O.; Roman, R.; Ambrus, R.; Zupko, I.; Moaca, E.A.; Coricovac, D.E.; et al. Cocrystal Formation of Betulinic Acid and Ascorbic Acid: Synthesis, Physico-Chemical Assessment, Antioxidant, and Antiproliferative Activity. Front. Chem. 2019, 7, doi:10.3389/fchem.2019.00092.
  2. Sathisaran, I.; Dalvi, S.V. Crystal Engineering of Curcumin with Salicylic Acid and Hydroxyquinol as Coformers. Cryst. Growth Des. 2017, 17, 3974–3988. [CrossRef]
  3. Lu, E.; Rodriguez-Hornedo, N.; Suryanarayanan, R. A rapid thermal method for cocrystal screening. CrystEngComm 2008, 10, 665–668. [CrossRef]
  4. Yamashita, H.; Hirakura, Y.; Yuda, M.; Teramura, T.; Terada, K. Detection of cocrystal formation based on binary phase diagrams using thermal analysis. Pharm. Res. 2013, 30, 70–80. [CrossRef] [PubMed]
  • In cocrystallization mass loss before melting temperature is of greater concerns for confirmation of solvated nature. Fraction of weight loss before strong degradation and after melting peak may be due to volatilization of components. We have different possibilities of cocrystal (AZT-PCM Dihydrate, AZT-PCM monohydrate and AZT-PCM. H2O1/2. CH3OH). When the TGA mass loss up to melting temperature is compared with the theoretically loss of possible cocrystals, it showed agreement with (AZT-PCM monohydrate). Though mass loss is greater than theoretical mass but lesser than all other possible forms.
  • Many cocrystals have shown little mass loss in TG thermogram after melting temperature which have not been observed in DSC.

Wang, C. et al. Enhancing Bioavailability of Dihydromyricetin through Inhibiting Precipitation of Soluble Cocrystals by a Crystallization Inhibitor. Cryst. Growth Des. 16, (2016).

de Almeida, A. C. et al. Mechanochemical synthesis, characterization and thermal study of new cocrystals of ciprofloxacin with pyrazinoic acid and p-aminobenzoic acid. J. Therm. Anal. Calorim.

We supposed semihydrated solvate cocrystal (AZT-PCM.H2O.1/2 CH3OH) where theoretical mass loss will be 3.64%. Because semihydrated solvate (AZT. H2O.1/2 CH3OH) is previously reported.

Was the physical mixture analysed by TGA and DSC? How different were the thermograms of the physical mixture in comparison to the ones of cocrystal?

  • The DSC of physical mixture (AZT and PCM) is previously reported in literature. The obtained cocrystal DSC is totally different from the reported physical mixture DSC. Because of this reason we did not go for measuring the DSC of physical mixture, however, we can repeat this experiment if the reviewer allows.

Maswadeh, H. Incompatibility of Paracetamol with Pediatric Suspensions Containing Amoxicillin, Azithromycin and Cefuroxime Axetil. Pharmacol. & Amp; Pharm. 2017, 08, doi:10.4236/pp.2017.811026.

Preparation of physical mixture is described as follows: “… equimolar amounts of both drugs were mixed well (for better homogeneity) and the resulting solid mass was then crushed gently…”.  This description is unclear. The mixing was in a vortex, a drum mixer, in a mortar. Why was the sample dried? Or performed by triplicate?

  • Typing mistakes are corrected and the procedure has been made clear, it is hoped that now it is easily understandable. The sample was dried to evaporate possible moisture absorbed from environment during the process of mixing and grinding.

Finally, although the links to DOI number are included, none of the references in the paper contain the page ranges as requested in the Instructions for authors. Please amend them.

  • Corrected accordingly in revised manuscript with thanks.

New authors are included. Or Acknowledgements have also been modified.

  • Worthy reviewer, yes by mistake they were not included in initial submission. We are sorry for that

Some typing or grammatical errors must be corrected (a physician describes instead of prescribes, azithromycin in figure 1, room temperature in figure 2 and 7, Vander Waals,. Unfinished sentence at the end of section 2.3.1, errors in the legend in Figure 8,.. Please revise carefully the document.

  • The manuscript was carefully revised and mistakes were corrected accordingly
  • Further, we have collected some results to justify our stance. The phase diagram, melting points of all the material measured at the same instrument and micrometric analysis have been carried out. The obtained results support the formation of cocrystals, these results have been included as supporting information. It is hoped that the reviewer will get agree with cocrystal formation.

Round 3

Reviewer 2 Report

I must insist that despite the literature reported by the authors to support their work, there are serious problems related to the PXRD technique in the manuscript:

  • The authors said that paracetamol monoclinic form I was used in this study. The simulated powder pattern (refcode HXACAN10) is compared to the one given by the authors and it does not match. Please revise it. Moreover, how is it possible that there are some peaks at so high angles (65 and 79º) for an organic compound? Not only in PCM but also in AZT DH some bumps are observed in the same area. Contamination? This is probably due to any inorganic impurity. Please revise that the sample was pure PCM and include the appropriate form used in the study.
  • Respect to the powder pattern corresponding to the cocrystal, these peaks at 65º and 79º are also observed. But also other from PCM such as at 25º, 38-39º, 45º. Moreover, the peak at ca 46º is surprisingly narrower than the rest. This pattern may correspond to a mixture. For this reason, it cannot be said “The diffraction pattern of cocrystal exhibited unique pattern compared to that of starting materials and their physical mixture. The unique PXRD pattern of cocrystal in comparison with its starting precursors indicate that new crystalline phase material was successfully formed exhibited crystalline phase”

Some articles are reported by the authors to show that their work is in line with those. But reading them carefully, it is possible to see in these articles big important differences respect to the manucript.

In this reference Cryst. Growth Des. 2020, 20, 3577−3583, the authors also describe the single crystal of the cocrystal. Moreover, the PXRD of bulk solid was compared with the simulated pattern from the single crystal.

In Cryst. Growth Des. 2016, 16, 5030−5039, no peaks of precursors are observed in the PXRD of cocrystals. Moreover, new peaks at low angles are observed confirming a bigger cell volume as may due to the formation of the cocrystal.

Journal of Thermal Analysis and Calorimetry (2020) 140, 2293–2303. Again new peaks at low angles appear (marked with arrows).

In Eur J Pharm Biopharm 2007;67:112–119, the crystal structure of this cocrystal had been previously described in Crystal Growth & Design 2003, 3, 6, 909–919. So the authors could compare the PXRD from the bulk solid with the crystal structure.

Finally, additional data regarding bulk and tapped densities, Carr’s index or Hausner ratio are included. But these techniques does not confirm the formation of a new cocrystal, these densities are regarded to a property of bulk solid and what it is more important, its preparation method.